# Gas6/TAM System: A Key Modulator of the Interplay between Inflammation and Fibrosis

**DOI:** 10.3390/ijms20205070

**Published:** 2019-10-12

**Authors:** Mattia Bellan, Micol Giulia Cittone, Stelvio Tonello, Cristina Rigamonti, Luigi Mario Castello, Francesco Gavelli, Mario Pirisi, Pier Paolo Sainaghi

**Affiliations:** 1Department of Translational Medicine, Università del Piemonte Orientale UPO, 28100 Novara, Italy; micolcittone@gmail.com (M.G.C.); stelvio.tonello@med.uniupo.it (S.T.); crigamo@tin.it (C.R.); luigi.castello@med.uniupo.it (L.M.C.); gavelli.francesco@gmail.com (F.G.); mario.pirisi@med.uniupo.it (M.P.); pierpaolo.sainaghi@med.uniupo.it (P.P.S.); 2Division of Internal Medicine, Immunorheumatology Unit, CAAD (Center for Translational Research on Autoimmune and Allergic Disease) “Maggiore della Carità” Hospital, 28100 Novara, Italy; 3IRCAD, Interdisciplinary Research Center of Autoimmune Diseases, 28100 Novara, Italy

**Keywords:** Gas6, Axl, TAM, MerTK, inflammation, fibrosis, cirrhosis, IPF

## Abstract

Fibrosis is the result of an overly abundant deposition of extracellular matrix (ECM) due to the fact of repetitive tissue injuries and/or dysregulation of the repair process. Fibrogenesis is a pathogenetic phenomenon which is involved in different chronic human diseases, accounting for a high burden of morbidity and mortality. Despite being triggered by different causative factors, fibrogenesis follows common pathways, the knowledge of which is, however, still unsatisfactory. This represents a significant limit for the development of effective antifibrotic drugs. In the present paper, we aimed to review the current evidence regarding the potential role played in fibrogenesis by growth arrest-specific 6 (Gas6) and its receptors Tyro3 protein tyrosine kinase (Tyro3), Axl receptor tyrosine kinase (Axl), and Mer tyrosine kinase protooncogene (MerTK) (TAM). Moreover, we aimed to review data about the pathogenetic role of this system in the development of different human diseases characterized by fibrosis. Finally, we aimed to explore the potential implications of these findings in diagnosis and treatment.

## 1. Introduction

Although extracellular matrix (ECM) deposition is a crucial process in wound healing, normal tissue repair can progress to severe fibrosis if the tissue injury is persistent or repetitive or if the wound-healing response itself becomes dysregulated [1]. Different triggers can lead to the development of progressive fibrotic diseases, such as infections, exposure to toxins/irritants/smoke, chronic autoimmune reactions, and ischemia [2]. The development of diffuse fibrosis impairs the physiological activities of involved tissues and organs, representing a common pathogenetic element of many chronic inflammatory diseases, such as liver cirrhosis, kidney disease, interstitial lung disease, autoimmune diseases, and heart failure [3]. This makes targeting fibrosis a potentially promising therapeutic strategy in different human conditions. However, despite its recognition as a relevant cause of morbidity and mortality worldwide, the mechanisms underlying fibrogenesis are still largely unknown; thus, there are few drugs specifically targeting fibrosis [4,5,6]. A deeper knowledge of the biological systems involved in fibrogenesis might contribute to the development of novel therapeutic tools. In this context, the protein growth arrest-specific 6 (Gas6) and its receptors represent a promising system for investigation.

## 2. The Gas6/TAM Receptors System

Growth arrest-specific 6 is a vitamin K-dependent protein, first identified in murine fibroblasts in 1988 [7]. The biological functions of Gas6 are mediated by the interaction with TAM receptors, one out of the twenty subfamilies of tyrosine kinase receptors. The TAM acronym is for the three members of this subfamily: Tyro3, Axl, and MerTK [8]. The system is highly pleiotropic, being involved in many different physiological functions, as previously reviewed [9].

All three types of TAM receptors can be found in human plasma in their soluble (sTyro3, sAxl e, sMer) forms as a result of membrane receptor cleavage mediated by two metalloproteases (ADAM 10 and ADAM 17); although a definite function of soluble receptors has not yet been defined, a decoy activity has been proposed [10,11]. Both Gas6 and the soluble form of its receptors have been tested as circulating biomarkers in different human diseases [12,13,14,15,16,17].

## 3. Gas6/TAM and Lung Fibrosis

Inflammation and fibrosis are key pathogenetic moments in the development of interstitial lung diseases (ILDs). The definition of “ILD” encompasses a heterogeneous group of diffuse parenchymal lung disorders, classified accordingly to specific clinical, radiological, and histopathological features [18]. Interstitial lung disease may develop subsequently to chronic exposure to toxic agents and drugs or as a consequence of lung involvement in systemic immune diseases (such as connective tissue diseases). Frequently, there is no evidence of a specific underlying cause and a diagnosis of idiopathic interstitial pneumonia (IIP) is therefore made. The most common form of IIP is idiopathic pulmonary fibrosis (IPF) [19], which is a lethal lung disease with a poor prognosis [20] and represents a challenging model of fibrosis. In fact, given that the mechanisms leading to ECM deposition in lung interstitium are unknown, the only potential therapeutic target is fibrosis per se. In 2014, the Food and Drug Administration (FDA) approved the use of the first antifibrotic agent for the treatment of IPF: pirfenidone inhibits transforming growth factor beta (TGF-β)-stimulated collagen synthesis, decreases the ECM, and blocks fibroblast proliferation in vitro; clinically, this attenuates disease progression, as reflected by lung function, exercise tolerance, and progression-free survival [21]. Even more interestingly, for the purpose of the present review, the FDA also approved the use of nintedanib for the treatment of IPF [22]. Since this molecule directly targets tyrosine kinase receptors, its effectiveness in IPF represents a potential rationale for the development of other drugs with a similar mechanism of action. However, it should be acknowledged that nintedanib has a wider activity, also targeting non-tyrosine kinase receptors. It is noteworthy that nintedanib is also effective in connective tissue diseases related to ILD, dampening fibrosis development independently from the underlying trigger [23].

Specifically looking at GAS6/TAM receptors, Fujino et al. first demonstrated that Axl-dependent signaling acts as a negative regulator of alveolar epithelial phenotype and function; the inhibition of Axl kinase activity suppresses mesenchymal cell functional properties, promoting epithelial cell traits. Interestingly, the activation of Axl is observed in areas of human IPF lung undergoing tissue fibrosis [24]. A recent paper deepened these observations, shedding a new light on the involvement of this system in lung fibrosis. Espindola et al. [25], in fact, demonstrated that, in lung samples obtained from IPF patients, Gas6 and Axl transcriptional levels were significantly higher than in healthy controls. The staining revealed that phosphorylated Axl was detected in fibroblastic foci, its expression being higher in patients with a rapid disease course. In IPF and normal fibroblasts cultures, the use of a small molecule (R428) targeting TAM receptors was effective in reducing the protein expression of collagen 1 and of α smooth muscle actin. Moreover, R428 was effective at inhibiting invasive wound and chemotactic invasion of both normal and IPF fibroblasts in an in vitro assay. Finally, R428 promoted apoptosis and inhibited proliferation in human lung fibroblasts.

The authors also assessed the in vivo role of Gas6/TAM receptors using two experimental models of pulmonary fibrosis. In a humanized severe combined immunodeficiency (SCID)/beige mutation (Bg) model, after IPF fibroblast injection, both Gas6 and Axl transcripts significantly increased in mice lungs, when compared with normal fibroblasts injected and non-humanized, naive, SCID/Bg mouse lungs. Moreover, R428 significantly decreased hydroxyproline content and phosphorylated Axl levels in mice lungs. In a bleomycin-induced pulmonary fibrosis model, *Gas6 −/−* knock out (KO) led to a significant decrease in hydroxyproline content when compared with wild-type mice at Day 14 after bleomycin exposure. Consistent with these findings, it has been recently reported that the genetic loss of Gas6 and MerTK mitigates the inflammatory response after silica intratracheal administration in a mouse model of silicosis. This was paralleled by a reduction of the collagen deposition [26].

It should be taken into account that the effect of Gas6/TAM receptors in the lung might be modulated by Protein S (ProS); in fact, ProS concentrations are lower in patients affected by IPF and non-specific interstitial pneumonia than in healthy controls. Moreover, in mice models, the transgenic overexpression of human ProS is protective against lung fibrosis, and the exogenous administration of human ProS ameliorates bleomycin-induced lung fibrosis [27]. It can be argued, therefore, that the evaluation of the potential role of the Gas6/TAM system in lung fibrosis should take into account the beneficial counterbalancing effect of ProS. Beyond these speculative considerations, all these very recent findings point towards a putative involvement of the Gas6/TAM system in the development of lung fibrosis; this hypothesis deserves further investigations, particularly considering the potential use of compounds targeting Gas6/TAM as a novel therapeutic strategy.

## 4. Gas6/TAM and Liver Fibrosis

Liver fibrosis and its final evolution, cirrhosis, are the result of tissue repair following chronic liver injury; indeed, though many different factors (viral, toxic, genetic, nutritional, etc.) can trigger liver damage, common pro-inflammatory and pro-fibrotic pathways lead to necrosis of hepatocytes, replacement of liver parenchyma by fibrotic tissue, and regenerative nodules with loss of liver function [28].

Interestingly, Gas6 plasma levels increase in response to acute liver injury in mice. In the liver, the repair process after an acute injury requires the recruitment of hepatic stellate cells (HSCs), which progressively change their phenotype towards myofibroblasts (HSCs/Myofibroblastic cells (MFBs)), releasing ECM. After CCl_4_-induced liver injury, the expression of Gas6 and Axl is upregulated in HSCs and macrophages populating damaged areas. The same authors demonstrated that in vitro Gas6 exerts an antiapoptotic effect on HSCs [29]. Consistently, *Gas6 −/− KO* mice are characterized by a defective wound healing after CCl_4_-induced liver damage, with reduced Kupffer cell activation and decreased macrophage and HSCs/MFBs recruitment in damaged areas [29,30]. In a further model, *Gas6 −/− KO* mice showed a massive hepatocellular damage after ischemia/reperfusion-induced injury. The administration of recombinant Gas6 has a protective effect on hypoxic damage in *Gas6 −/− KO* mice, which seems to be mediated by the interaction of the ligand with MerTK [31]. When cultured and stimulated with lipopolysaccharide, MerTK-expressing monocytes show a reduced production of proinflammatory cytokines, reversed by the use of a MerTK inhibitor [32]. Consistently, *MerTK −/− KO* mice exhibit persistent liver injury and inflammation after acetaminophen (APAP)-induced acute liver damage [33]. The anti-inflammatory activity of Gas6 is confirmed by the observation that triple *TAM −/− KO* mice show a phenotype which clinically and histologically resembles autoimmune hepatitis, with persistent infiltration of inflammatory cells and elevation of pro-inflammatory cytokines [34]. An increased number of monocytes and macrophages expressing MerTK in the circulation, liver, and lymph nodes, compared with patients with stable cirrhosis and controls, has also been described in human subjects with acute or chronic liver failure. [32].

Fibrosis is a physiological response to tissue damage; therefore, it is reasonable to postulate that this protective effect of Gas6 and its receptors may be played at the expense of the development of fibrosis and scarring. In fact, when compared to WT mice, knocking out Gas6 myofibroblast activation in a murine model of steatohepatitis leads to a reduction of the hepatic expression of TGF-β, collagen 1, and collagen 3 mRNA and prevents the development of liver fibrosis. Interestingly, this protective effect is not specifically associated to a causal factor, being *Gas6 −/− KO* mice protected against liver fibrosis also in case of repetitive CCl_4_ exposure [35]. In the pro-fibrotic activity of Gas6, both MerTK and Axl seem to be relevant. As previously mentioned, Axl expression is upregulated after CCl_4_ challenge in mice; the crucial role of Axl is confirmed by the observation that *Axl −/− KO* mice are protected against liver fibrosis after toxic injury. As a further clue, Axl is required for HSC activation in vitro, and HSCs from *Axl −/− KO* mice express significantly lower levels of α-SMA and Col1a1 transcripts [36]. Similarly, HSCs express MerTK and its stimulation with Gas6 leads to an increase of procollagen 1 expression, which is blunted after MerTK silencing [37,38].

The involvement of MerTK in chronic liver diseases (CLDs) in humans has been explored by a genome-wide association study, which reported a linkage between the non-coding single nucleotide polymorphism (SNP) rs4374383 G > A of the *MERTK* gene and the risk of liver fibrosis progression in hepatitis C virus (HCV) patients [39]. This observation was confirmed in a longitudinal study, which agreed in identifying the A/A homozygosity as protective for liver fibrosis progression in HCV-related CLD [40]. Similarly, this genotype is protective in non-alcoholic fatty liver disease (NAFLD): A/A homozygosity was associated to a lower rate of significant fibrosis and to a lower liver expression of MerTK.

These findings prompt two considerations: (i) being increased in case of liver damage, Gas6 is a putative biomarker of CLD; (ii) being associated to development of liver fibrosis, targeting Gas6/TAM axis might be considered a potential therapeutic strategy in CLD.

In 2015, Bárcena et al. first tested Gas6 and sAxl as biomarkers of CLD. They observed an increase in Gas6 and sAxl plasma levels in patients affected by alcohol-related cirrhosis with respect to controls, showing close correlation with the severity of the disease. They also tested these biomarkers in HCV-related CLD. Accordingly to their data, Gas6 levels were significantly different between individuals with established fibrosis (F2) and patients with initial fibrosis (F0 and F1 groups), while sAxl levels displayed significant changes between patients with F2 fibrosis and individuals with advanced fibrosis or cirrhosis (F3/F4 group) [36]. More recently, our group [41] demonstrated that Gas6 plasma concentrations increase for increasing liver stiffness in patients affected by CLD of different causes. In the subgroup of patients undergoing liver biopsy, those with a significant fibrosis showed higher Gas6 levels, than those with absent/minimal fibrosis. Finally, the diagnostic accuracy of Gas6 was comparable to that of liver elastography. Notably, Gas6 accurately performed also in the identification of esophageal varices in a population of HCV-related CLD [42]. Similar data have been reported on sAxl [43]. Consistent with the report by Barcena et al. [31], sAxl predicts advanced liver fibrosis, being significantly higher in patients with cirrhosis than in those without. This observation is in line with the results of a large multicenter study, according to which sAxl levels are higher in patients with advanced liver fibrosis (cirrhosis or histologically defined F4) than in controls, independent from the underlying etiology. Moreover, sAxl is a biomarker of hepatocellular carcinoma either when identified in cirrhotic or in non-cirrhotic liver [44].

To the best of our knowledge, there are no in vivo data about targeting TAM receptors to suppress liver fibrosis development; the antifibrotic effect of Axl [36] and MerTK inhibition [37] has been explored with promising results in vitro.

## 5. Gas6/TAM and Cardiovascular Remodeling

Cardiovascular remodeling is an adaptive process caused by different injuries (inflammation, ischemia, ischemia/reperfusion (I/R), biomechanical stress, excess neurohormonal activation) [45]; although protective towards local damage, the ongoing process contributes to different conditions, such as contractile dysfunction, atherosclerosis, and hypertension, finally participating in the pathogenesis of cardiovascular diseases (CVDs).

Once more, the Gas6/TAM system exerts a protective role against the inflammatory process following acute local damage. The role of MerTK is crucial in favoring cardiac remodeling in mice, after an ischemic injury [46]; notably, *MerTK −/− KO* mice show larger infarction size after ischemic injury than WT mice, caused by a defective clearance of apoptotic bodies. In fact, MerTK deficiency led to an accumulation of apoptotic cardiomyocytes and a reduced index of in vivo efferocytosis. Suppressed efferocytosis preceded increases in myocardial infarct size and led to delayed inflammation resolution and reduced systolic performance [47]. Moreover, cultured splenocytes isolated from *MerTK −/− KO* mice showed enhanced production of pro-inflammatory cytokines and MerTK deficiency led to accelerated atherosclerosis [48]. Conversely, mice expressing cleavage-resistant MerTK showed increased levels of specialized pro-resolving mediators such as TGF-β and IL-10 [49]. It should be underlined, however, that TGF-β is a crucial growth factor in fibrogenesis. This further suggests that the anti-inflammatory activity of Gas6/TAM is paralleled by fibrogenesis enhancement.

Cardiovascular remodeling is mainly mediated by vascular smooth muscle cells (VSMCs). Growth arrest-specific 6 is secreted by VSMCs and enhances their proliferation [50], probably via Axl activation; in fact, in rats, an acute vascular injury induces Axl expression in VSMCs [51] and *Axl −/− KO* reduces intimal thickening following vascular injury [52]. Thus, Gas6/Axl seems to play a crucial role in cardiovascular remodeling by induction of proliferation, migration, and protection from apoptosis of VSMCs [53].

The Gas6/Axl system has a role in cardiac remodeling. The *GAS6 −/− KO*’s mice have normal left ventricular structure and function, but after aortic banding develop less hypertrophy, fibrosis, and contractile dysfunction when compared with WT mice. Conversely, cardiac-specific overexpression of GAS6 exacerbated aortic banding-induced cardiac hypertrophy, fibrosis, and dysfunction [54]. This effect is probably mediated by Axl, which has been reported to be expressed by cardiomyocytes [55]. Interestingly, in patients with heart failure, cardiac expression of Axl is enhanced, as shown on myocardial biopsies of end-stage patients undergoing transplant compared to donors. The serum concentration of the sAxl increased as well, being predictive of several major heart failure events (all-cause mortality, heart transplantation, and hospitalizations) at one-year follow-up [56].

## 6. Conclusions

In the last few years, there has been a growing interest in the potential role of Gas6 and its receptors in human diseases. Different reviews have been published, particularly regarding Gas6’s involvement in the development of liver fibrosis [57,58]. However, in the present paper we evaluated a novel aspect of the Gas6/TAM system: its bivalent role in the modulation of the interplay between inflammation and fibrogenesis.

In light of the current literature, it is evident that the Gas6/TAM system has anti-inflammatory properties, mainly deriving from the modulation of macrophage activity. Both receptors and ligands are overexpressed by damaged tissues. Here, Gas6 exerts its anti-inflammatory activities, suppressing the production of proinflammatory cytokines and mediating efferocytosis of apoptotic bodies, thus limiting potential antigens available for antigen presenting cells (APCs), attenuating the signal transduction of Toll-like receptor (TLR) and type I Interferon IFN, and inhibiting NLRP3 inflammasome activation by autophagy [59]. All these outcomes are desirable in the case of acute injury. Consistently knocking-out Gas6 or TAM receptors increases susceptibility to acute stressors, as demonstrated in the liver and heart of murine models. But the response to damage invariably requires reparative processes, which are of benefit when limited and becoming pathogenic per se when excessive. In fact, the development of fibrosis in response to tissue damage is a common pathogenetic moment of different human diseases. Noticeably, despite different etiologies, this process always follows common pathways.

The protective effect of Gas6 on human tissues contemplates the activation of fibrosis; in fact, in both in vitro and in animal models, this protein shows profibrotic properties which probably involves interaction with both MerTK and Axl. It can be deduced that, in the case of chronic tissue damage, the advantage derived by its beneficial effect on inflammation is paralleled by a fibrogenic activity, characterized by deposition of ECM, progressive parenchymal derangement, and, finally, loss of function. This is why, when mice are exposed to chronic tissue damage, the blockade of the system, by knocking out genes or by the use of selective inhibitors, prevents the development of fibrosis, as reported in the liver, lungs, and heart. This dual effect on acute and chronic injury has already been proposed as an explanation for Gas6/TAM actions in the immunopathogenesis of liver disorders, but is probably a valid model for other human organs [60] (also see Figure 1). Besides the growing evidence linking this system to lung and cardiovascular diseases, it should be acknowledged that clinical data and experimental models suggest a potential involvement in kidney inflammation and sclerosis, although the evidence on this topic is more controversial [61,62].

On this basis, it is not surprising that targeting the TAM system is emerging as a novel field of investigation for the treatment of human diseases, although its potential application should be weighed in light of its beneficial effect on inflammation. However, more studies are required to test the efficacy and safety of TAM targeting as a novel strategy for fibrosis treatment.

## Figures and Tables

**Figure 1 ijms-20-05070-f001:**
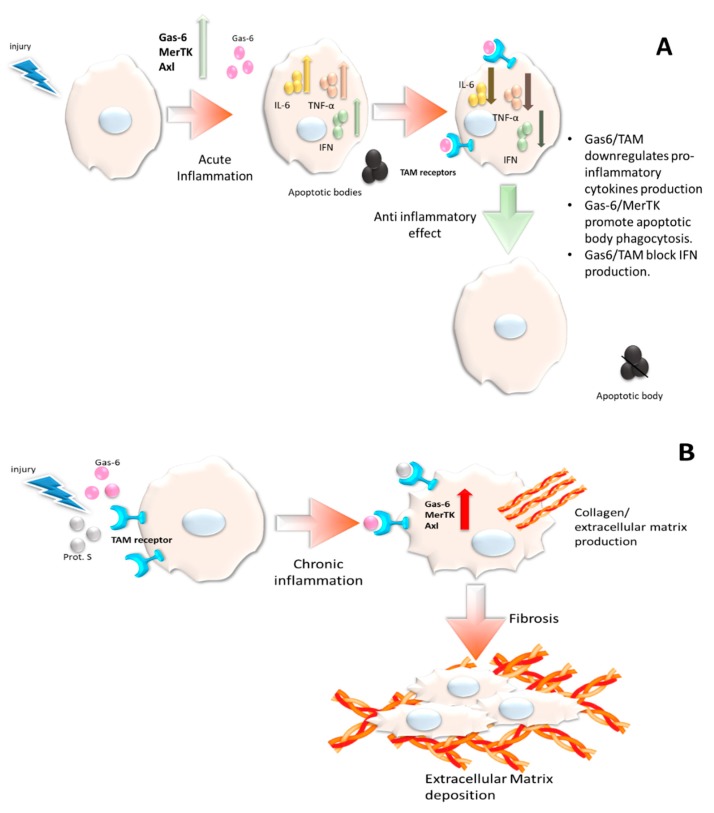
A potential model of Gas6/TAM activity in the case of acute and chronic injury. (**A**) A cell exposed to a damage overexpressing Gas6/TAM receptors. The arrow pointing upward means increasing. Down arrow means decreasing. The system exerts its anti-inflammatory activities: efferocytosis, switch-off of IFN signature, and downregulation of inflammatory cytokines expression. This contributes to self-limitation of inflammatory response. (**B**) The overexpression of Gas6/TAM receptors induced by chronic exposition to a stressor, leading to the production and release of extracellular matrix, ultimately contributing to fibrosis development.

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
