# Peer review of "Gas6/TAM System: A Key Modulator of the Interplay between Inflammation and Fibrosis"

_ijms, 2019, doi:10.3390/ijms20205070_

Round 1

Reviewer 1 Report

In this paper the authors discussed the role of Gas6 and its receptors in fibrosis . The manuscript is clearly written. The structure of manuscript was properly designed with appropriate balance between the sections. However, I have found some points in the manuscripts that may be improved to increase the value of work. Please find my comments in authors section.

Main concerns

I suggest to remove or replace section 2, i.e. ,,Gas6/TAM receptors system’’ The same information was included by authors in their previous publication: Bellan M, Pirisi M, Sainaghi PP. The Gas6/TAM System and Multiple Sclerosis. Int J Mol Sci. 2016;17(11):1807. Published 2016 Oct 28. doi:10.3390/ijms17111807. Rearrange this section to present new information or just cite above mentioned publication, please.

Minor comments:

Please check authors’ contribution section, methodology or data curation in review ?

Author Response

Dear Editor,

first of all I would like to thank you and the reviewers for the valuable comments; please find attached a rebuttal to Reviewer’s 1 observations:

Main concerns:

We have replaced section 2, limiting it to the fundamental notions to introduce Gas6/TAM system.

Minor comments:

Sorry for the mistake; we have rephrased author’s contributions.

Best regards

Reviewer 2 Report

The review by Bellan et al. summarizes the evidence of the implication of the TAM receptor tyrosine kinases and their ligands in the process of pathological fibrosis. The authors propose molecules of the system as adequate markers of the fibrotic process and therefore as valuable diagnostic tools. Besides, they conclude that the system could be an interesting target for therapy.

The subject of the review overlaps with similar review articles published recently, including Holstein et al (IJMS 2018); Smirne et al (Disease Markers 2019) and Bellan et al (J Clin Transl Hepatol 2018), the last two by the same group of the authors. However, the present review is the first to include the potential role of the TAM system in fibrosis in other organs besides the liver. The authors should indicate the differences of the present review with other recent reviews in the field in order to stress its novelty.

In general, the review is well written and summarizes the main studies in this promising field of research. Nevertheless, I would suggest that the authors include recent reports on the role of TAMs in kidney fibrosis.

There are several specific problems in the text.

Line 47: use residues instead of “residuals”. Glu residues in the GLA module are potentially gamma carboxylated. Most molecules are partially gamma-carboxylated. I suggest to change the sentence to “…contains 11 potential…”.

Line 106. “…in an vitro assays.” Correct.

Line 119. In line 44 protein S is abbreviated ProS. This nomenclature should be used throughout the manuscript, instead of protein S (L119, L121, L122, L124).

Line 130 to L134. I do not follow the meaning of this sentence. Could you check the grammar?

L136. “reqquires”.

L149. From reference 66, the reference list seems wrong (see note below).

L168. Could the authors clarify what do they mean by “exploited” in this context?

L176 CLD should be explained in the first appearance.

L208 “Mertk” change to “MerTK”

Line 223 Gas/Axl should be “Gas6/Axl”.

L228 to L231. The references to these studies are missing.

L464-L466. Ref 66 and 67 seem to refer to the same article. From this point the reference list seems wrong, as the reference in the text seem to use the following number (i.e. 81 in the text reference to reference 82 in the reference list).

Author Response

Dear Editor,

first of all I would like to thank you and the reviewers for the valuable comments; please find attached a rebuttal to Reviewer’s 2 observations:

We have highlighted the the differences of the present review with previous papers dealing with this topic, as requested. We have added a comment dealing with Gas/TAM system and kidney inflammation and fibrosis We have corrected the specific issues highlighted by the reviewer.

Best regards

Reviewer 3 Report

This review article provides the information related to the roles of Gas6/TAM in the inflammation and fibrosis. There are only minor concerns/suggestions (list below).

Minor concerns:

(1) Lines 30-32: “Different triggers can lead to the development of progressive fibrotic diseases; among them, for instance, we should list infections,….”

This is the only one sentence using the subject “we” in the entire article. To make a smooth transition, “, such as infections,….” can be used to replace “; among them, for instance, we should list infections,…….”  

(2) Lines 91-93: The drug, nintedanib, not only targets tyrosine kinase receptors (mainly), but also non-tyrosine kinase ones, which should also be considered.

(3) Line 168 vs Line 176: The “chronic liver disease (CLD)” appeared in Line 176, whereas the abbreviation “CLD” was already used in Line 168. These orders should be reversed.

(4) Similarly, in the texts, some diseases and/or drug names only appeared as abbreviations, but some had both abbreviations and full names listed. The authors should carefully check these to ensure all abbreviations displayed consistently.

For example, the full name of “APAP” (in Line 147, APAP-induced acute liver damage..) was never incorporated in the text. It’s only shown in the abbreviation list.

(5) The list of Abbreviations should be revised. The Abbreviations are not listed in the alphabetical orders, nor in the chronical orders completely. If the authors want to use the chronical order as each abbreviation appearing in the text, then footnotes should be used.

Author Response

Dear Editor,

first of all I would like to thank you and the reviewers for the valuable comments; please find attached a rebuttal to Reviewer’s 3 observations:

Minor concernss:

Rephrased as suggested; We acknowledged in the text the wider activity of nintedanib on non-tyr kinase receptors We have inverted the order as suggested; We added the missing full names We have re-ordered the list in alphabetical order

Best regards